# IoTwins: Toward Implementation of Distributed Digital Twins in Industry 4.0 Settings

Alessandro Costantini [1,*], Giuseppe Di Modica [2], Jean Christian Ahouangonou [3], Doina Cristina Duma [1], Barbara Martelli [1], Matteo Galletti [1], Marica Antonacci [4], Daniel Nehls [5], Paolo Bellavista [2], Cedric Delamarre [3] and Daniele Cesini [1]

1  Center for Research and Development on Information and Communication Technologies (CNAF), Italian Institute for Nuclear Physics (INFN), 40127 Bologna, Italy; cristina.aiftimiei@cnaf.infn.it (D.C.D.); barbara.martelli@cnaf.infn.it (B.M.); mgalletti@cnaf.infn.it (M.G.); daniele.cesini@cnaf.infn.it (D.C.)
2  Department of Computer Science and Engineering, University of Bologna, 40132 Bologna, Italy; giuseppe.dimodica@unibo.it (G.D.M.); paolo.bellavista@unibo.it (P.B.)
3  ESI GROUP, 94150 Rungis, France; jean-christian.ahouangonou@esi-group.com (J.C.A.); cedric.delamarre@esi-group.com (C.D.)
4  Italian Institute for Nuclear Physics (INFN) Sez. Bari, 70126 Bari, Italy; marica.antonacci@ba.infn.it
5  Fraunhofer FOKUS, 10589 Berlin, Germany; daniel.nehls@fokus.fraunhofer.de
*  Correspondence: alessandro.costantini@cnaf.infn.it

**Abstract:** While the digital twins paradigm has attracted the interest of several research communities over the past twenty years, it has also gained ground recently in industrial environments, where mature technologies such as cloud, edge and IoT promise to enable the cost-effective implementation of digital twins. In the industrial manufacturing field, a digital model refers to a virtual representation of a physical product or process that integrates data taken from various sources, such as application program interface (API) data, historical data, embedded sensor data and open data, and that is capable of providing manufacturers with unprecedented insights into the product's expected performance or the defects that may cause malfunctions. The EU-funded IoTwins project aims to build a solid platform that manufacturers can access to develop hybrid digital twins (DTs) of their assets, deploy them as close to the data origin as possible (on IoT gateway or on edge nodes) and take advantage of cloud-based resources to off-load intensive computational tasks such as, e.g., big data analytics and machine learning (ML) model training. In this paper, we present the main research goals of the IoTwins project and discuss its reference architecture, platform functionalities and building components. Finally, we discuss an industry-related use case that showcases how manufacturers can leverage the potential of the IoTwins platform to develop and execute distributed DTs for the the predictive-maintenance purpose.

**Keywords:** cloud; edge; IoT; digital models; digital twin

## 1. Introduction

The fourth industrial revolution, better known as *Industry 4.0* (I4.0), promotes a significant transformation of the industrial manufacturing by means of the extensive digitization of factories. I4.0 is grounded on the all-connected factory concept, according to which the shop floor and the office floor departments, historically isolated from each other, integrate to form a continuum of digitized assets that cooperate to optimize the production process and, overall, to grow the business. Worldwide, several initiatives have proposed standards and specifications to support the digitization goals set by I4.0 [1–3]. Several information technologies (ITs) are recognized by practitioners as the "enablers" of the aforementioned revolution. Among those, the industrial Internet of Things (IIoT) and edge and cloud computing are definitely the ones that gained the most penetration in industrial settings.

The strong interest in the adoption of these technologies in the industrial manufacturing field is also witnessed by the many projects that the European Commission has funded in the context of the Horizon 2020 Research and Innovation Programme. Moreover, The European Open Science Cloud Hub [4] contributed to the development of a Hub for European researchers and innovators to discover, access, use and reuse a broad spectrum of resources for advanced data-driven research. The possibility for companies, in particular, small and medium enterprises (SMEs), to access those investments has strengthened the collaboration among public and private research.

In such respects, the EU-funded IoTwins project [5] aims to develop a SME-affordable platform to build innovative solutions for the optimization of production processes and the smart maintenance of assets in both the manufacturing and the facility management sectors. Among the 12 testbeds supported by the project, 4 come from the manufacturing sector and are experimenting with innovative predictive diagnosis/maintenance solutions in several domains, namely, wind turbines, machine tools for the production of automotive components, machines for the production of crankshafts and machines for the production of bottle caps. Facilities and infrastructure management is investigated by three testbeds: focusing on the optimization of crowd management, on the smartification of an electrical grid and on the power optimization of supercomputing infrastructures. The replicability and standardization of previous models are covered by another five testbeds aimed at defining new areas of application and developing innovative business models in the manufacturing and facility domains.

The IoTwins project leverages edge-enabled and cloud-based big data services to support the design and development of digital twins (DTs) to be deployed in the IoT–edge–cloud continuum in order to achieve the aforementioned goals. DTs are virtual copies of a company's assets that can interact with each other and, by leveraging the knowledge extracted by large amounts of data, implement corrective actions, optimize efficiency and diagnose anomalies before they occur. The project focuses on *hybrid DTs*, i.e., DTs that mix a data-driven and a model-driven approach. The data-driven component of the hybrid DT is a machine learning (ML) model that is trained in the cloud with historical and real data coming from the field, and is then executed on the edge (i.e., close to the field) to ensure prompt reactions to anomalous situations. The model-driven component is a software simulator that mimics the behaviour of physical assets and can also provide useful information to the data-driven model.

In the course of the project, a distributed platform and a number of cloud- and edge-enabled DTs have been designed, developed and made available to the project stakeholders. The mentioned project's output will be released to the research and industrial community after the end of the project. In this paper, we introduce the software platform developed in the context of the IoTwins project and provide some technical details of its implementation. We also present and discuss a manufacturing use case, developed by one of the project partners, that demonstrates the practical benefits of the proposed solution. This work is grounded on the research results published in [6], and advances them by (a) proposing further technical details of the platform and (b) showcasing the platform potential in a real manufacturing scenario.

The paper is structured in the following way. The related work is drafted in Section 2. In Section 3, we introduce the use cases-driven approach adopted in the IoTwins project. In Section 4, we disclose some details of the platform design, while specific platform capabilities are discussed in Section 5. In Section 6, we discuss the experience of a manufacturing company that implemented DTs with the support of the platform tools. Section 7 drafts the final conclusions and touches on the future activities.

## 2. Related Work

In the literature, there is a growing interest around I4.0 and the benefits that it will bring at present and in the near future. The research communities are addressing the many facets of the revolution and focus on the technologies that will speed up manufacturing

enterprises' transition to digital [7,8]. The authors of [9] survey the technologies and domains affected by the Industry 4.0 transition. They discuss the emerging trend of merging and interconnecting information and communication technologies (ICTs), cyber-physical systems (CPSs) and the IoT in modern production plant deployments. Ref. [10] proposes the I4.0 vision under the perspective of a seamless integration of physical industrial resources and IT processes that will bring efficiency and increase the profits of plants.

In the transition to Industry 4.0, switching to digital approaches is mandatory in order for the operational technology of manufacturing industries to embrace software-defined approaches [11]. A key concept in this transition is *digital twins*, which are virtual representations of physical assets [12,13]. Digital twins facilitate the administration of physical assets and help designers build a digital model of the physical characteristics of the asset, so that it can be transferred or replicated on computing nodes [14]. Collaborative environments and efforts to streamline the application of digital twins are the focus for authoritative bodies such as SPARTA [15] in the field of cybersecurity, AI4EU [16], unifying the European Union infrastructure and framework for AI advances and Fortissimo [17], a collaborative project that enables European SMEs to be more competitive globally through the use of advanced modeling.

Digital twins have also been addressed by the major cloud providers, e.g., Amazon Web Services (AWS) [18], Microsoft Azure [19] and Google Cloud Platform (GCP) [20]. For example, within AWS' IoT, JSON representations of the real "thing" are used as devices shadows, which basically contain a *desired* and a *reported* state.

Leveraging computing resources provided in the cloud continuum, distributed digital twins may be employed that span the cloud–fog–edge chain. Fog/edge-assisted digital twins are best suite to support the management of assets involved in time-critical operations. Edge and fog computing were the focus of projects such as AUTOWARE [21], bringing fog technlogy into the manufacturing industry, BEinCPPS [22], which enables real-time machine-to-machine communication and Boost 4.0 [23], a European initiative that applies big data in manufacturing applications through fog/edge technology.

Cloud–edge architectures can be created by using AWS' IoT Greengrass [24]. Here, smaller devices may not connect directly to the cloud, but to a dedicated Greengrass core (GGC) device located on the edge that is able to act locally on the sensed data [25]. In this architecture, the GGC is the only device that needs a connection to the cloud. AWS' IoT SiteWise [26] is a managed service provided to collect, store, organize, and monitor data from industrial facilities at a large scale. Models of the physical assets, processes and facilities are used to provide metrics, e.g., for the prediction of maintenance issues.

The IoTwins project proposes a highly distributed (cloud-to-things) and *hybrid* digital twin model, which provides for an integration of simulative and data-driven models to feed AI services. To the best of our knowledge, this approach has not yet been explored extensively in the literature. The project can rely on the outcomes and the experience of past EU-funded projects such as: (i) the DEEP Hybrid DataCloud (DEEP-HDC) project [27], which aimed to bridge together cloud and intensive computing resources in order to explore different datasets for artificial intelligence and deep and machine learning; (ii) the eXtreme-DataCloud (XDC) project [28], which focused on developing scalable technologies aimed at federating storage resources and managing data in highly distributed computing environments; (iii) the INDIGO-DataCloud project [29], which centered on developing a computing platform that can be deployed on different hardware and provisioned over hybrid infrastructures.

## 3. IoTwins: A Use Case-Driven Project

The IoTwins project aims at democratizing Industry 4.0 for companies that do not have enough resources to invest in these tools. That is why IoTwins specifically targets SMEs in the corresponding sectors (automotive components production, energy production, goods) and in the service sector (IT infrastructure, sport facilities, smart grids) to lower the barriers for the use of DTs and increase productivity, safety and resiliency.

The IoTwins project was conceived to cater to the digitization needs of business companies in the aforementioned sectors. In the project, much effort was devoted to the elicitation of user requirements (that made IoTwins a "use case-driven project") and to the design of easy-to-use tools that would help them to exploit the potential of technologies, such as edge/cloud computing and AI, transparently and efficiently.

In IoTwins, the user is the principal stakeholder that will benefit from the project results. In the scope of the project, the IoTwins user role is played by so-called *testbed owners*, i.e., business companies participating in the project as piloting partners whose contribution includes providing requirements and developing a DT-enabled testbed. Out of the project scope, potential users are players from the manufacturing and facility/infrastructure management sectors that intend to benefit from the IoTwins platform services to implement DT-based management of their business processes.

At the beginning of the project, testbed owners ("users", from now on) had to take extensive surveys from which a clear picture of technological needs was drawn. To build distributed DTs that are capable of exploiting the computing resources offered in the IoT–edge–cloud computing continuum, users needed a simple mechanism that allowed them to (i) pick atomic services from a list of ready-to-use components, (ii) wire services to form a service chain (i.e., the DT), (iii) deploy the chain on the computing continuum and, finally, (iv) run it. Therefore, from the user's perspective, DTs should be implemented following the *pick'n'compose* approach, which provides for an easy build of a complex service by simply selecting, composing and configuring its building blocks.

To meet the mentioned needs, the IoTwins platform will offers users services to select, configure and run data processing tasks on the IoT, edge and cloud, respectively (more details about the components are given in the following Sections). To do so, the use cases proposed by the partners have been categorized according to the infrastructure where computing tasks are run; the use case flow is described hereafter for each referenced infrastructure (see Table 1 for details).

**Table 1.** Use case flow and related computational tasks mapped for the IoTwins infrastructure.

| Referenced Infrastructure | Use Case Flow | Data Processing |
|---|---|---|
| IoT | Configure and run data processing. | On-the-fly data processing or data processing at rest. |
| Edge | Configure data processing, run bulk data and data stream processing, run ML model. | Data stream processing. |
| Cloud | Configure data processing, run bulk-data and data stream processing, run ML training and simulation. | Data streams from either IoT or edge or data-at-rest stored in the cloud data processing. |

Under the "data processing" umbrella, all activities and tasks pertaining to data analytics (data filtering, data polishing, data integration, data elaboration, data visualization, data monitoring) are covered. In Table 1, use cases are categorized according to the infrastructure where computing tasks are run.

The category "IoT use cases" includes the configure and run data processing. These refer to the action of configuring and running computing tasks on data sensed by sensors that the IoT device is equipped with. Depending on the need and the capabilities of the device, computing tasks may consume data on the fly or consume data at rest. The computing tasks that are expected to run depend on the constraints that characterize IoT devices and on the data preparation tasks (e.g., data cleaning, data pre-filtering, etc.).

The "edge use cases" include: configure data processing, run bulk data processing, run data stream processing and run ML model. Through the configure data processing task,

the user is allowed to set up an environment customized for data elaboration. Moreover, configuration involves also actions such as (a) setting up the data sources' address, type and format, (b) defining and setting parameters for a specific computing task, (c) selecting the destination of the data output, etc. The run bulk data use case, instead, invokes tasks able to elaborate data at rest, i.e., locally stored data. The run data stream processing use case refers to the elaboration of live data coming from IoT devices. Finally, the run ML model use case is a sample data stream processing use case.

The "cloud use cases" include the same actions described for the "edge use cases", but in this case, the user can configure and run computing tasks on data streams coming from either IoT or edge devices, or on data-at-rest stored in the cloud. As an example, the user can request to run simulations (run simulation use case) or to train ML models (run ML training case) on bulk data present in the cloud. Trained ML models can then be moved to the edge where, fed by data streams coming from IoT devices, they will execute (see edge's "run ML model" use case).

Finally, use cases have been defined to let the user request the activation of multiple tasks distributed along the chain of the computing infrastructures. The user can set up a data processing chain (configure data processing chain use case) and run it (run data processing chain use case). Depending on the needs, chains can be configured to span two infrastructures (IoT–cloud, IoT–edge) or all infrastructure levels (IoT–edge–cloud). As an explanatory example, a typical data processing chain envisions data sensing and filtering tasks deployed in the IoT, an ML model training task running in the cloud and fed with data at rest and an ML model execution task, running in the edge and consuming IoT data streams. Along the chain, data transfer services (bulk data transfer, live data transfer) and storage services (relational/NoSQL/time series DB, file-system based storage, etc.) can be instantiated.

According to the above-described needs, as from columns two and three of Table 1, the following functionalities have been designed and implemented into the IoTwins platform:

- Support data transfer to/from the three infrastructure levels in ways that account for the constraints imposed by the specific computation needs (real-time, non-real-time);
- Support several data types and accommodate different data storage needs (short- and long-term storage);
- Support the elaboration of data both on the fly (streamed-data elaboration) and at rest (batch-data elaboration).

Apart from offering the users support for building and running services on each of the computing infrastructure levels, the platform provides services to configure, build and transparently deploy complete "computing chains", i.e., chains of software libraries/tools/services that implement the above-mentioned functionalities along the IoT–edge–cloud continuum.

## 4. Design of an Open Platform for Digital Twins

The IoTwins platform has been proposed as an open framework that enable Users to develop, configure and run DTs in the cloud-to-things continuum. The platform is built making use of open-source software and tools. Moreover, third-party software have been integrated by using open APIs.

The proposed platform's architecture is drafted in Figure 1.

The figure shows the different computing layers targeted by the platform and the related interactions.

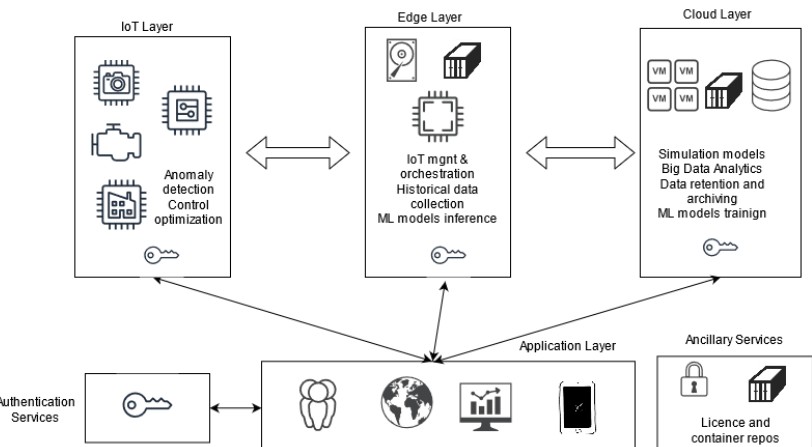

**Figure 1.** IoTwins' high-level architecture.

Openness and software re-use are the base concepts that drove the design of the IoTwins platform. As already mentioned, the technologies adopted to design and implement the IoTwins platform come from the requirements and the experiences collected in the previous cloud-related projects. The different components and tools are extensively presented in the next section, where the interactions among them are also described. By harnessing and integrating the functionalities of existing technologies and tools, the platform provides users with different services such as data handling, computation, encryption and anonymization:

- **Data handling**: this refers to the process of gathering, recording and presenting information in a way that is helpful to the final user. It includes three different consecutive steps: data gathering, data transportation and data storage. Data gathering is performed by the sensing devices from the field. Data transportation provides the protocols to implement communication among the three layers (IoT, edge, cloud). Currently, data transportation leverages standard (or de facto standard) technologies to maintain compatibility among the services. Data storage solutions (data models and databases) are used either as preliminary storage or as long-term storage for the testbeds;

- **Computation**: this is requested at all layers. At the IoT layer, a few computation tools are needed, such as those suited for light and quick calculations enabled on some types of devices. At the edge layer, filtering, data aggregation and big data analytics can be activated. At the cloud layer, heavy simulations and ML training are mainly carried out;

- **Anonymization**: this consists of replacing/obfuscating/removing data that can identify individuals, both directly and indirectly. This is an important issue for industry, where data need to be kept under strict confidentiality;

- **Encryption**: this is used to encrypt channels for data transfers. Due to the heavy computational cost of this service, it is provided in the edge and cloud layers.

## 5. IoTwins Platform's Components and Functionalities

The IoTwins platform offers a single point of access to heterogeneous computing, high capacity storage for heavy computation and a network of interconnected resources that cater to any computing need. The platform exposes interfaces to data analytics and AI techniques, physical simulations, optimizations and virtual lab services. In addition, it complies with well-established and emerging standards to enable communication- and data-level interoperability in the industrial IoT domain. In Figure 2, all the IoTwins service components are drawn together with the connections among them. Those components are explained in the following subsections.

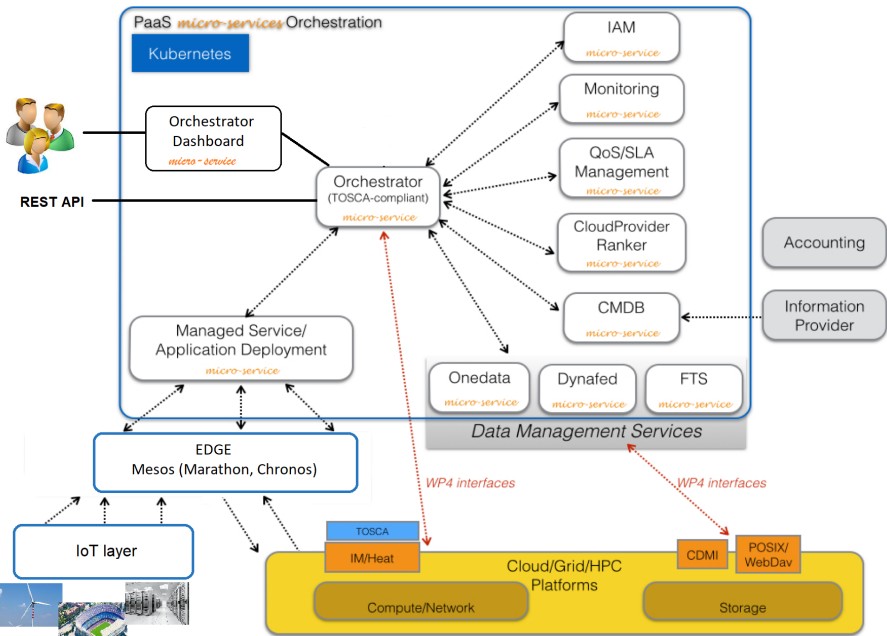

**Figure 2.** Graphic schema showing the connections among the IoTwins platforms' (INDIGO PaaS, edge and IoT) service components.

### 5.1. Authentication and Authorization

The authentication and authorization infrastructure service (AAI), among the above-mentioned stack of layers of the IoTwins platform, is provided by the INDIGO Identity and Access Management Service [30] (INDIGO-IAM).

Developed within the INDIGO-DataCloud EC project, the service is maintained by INFN and it is provided through multiple methods (SAML [31], OpenID Connect [32] and X.509 [33]) by leveraging the credentials provided by the existing identity federations (i.e., IDEM [34], eduGAIN [35], etc.). The selected access to the resources, as well as data protection and privacy, is provided by the support to distributed authorization policies and the Token Translation Service.

INDIGO-IAM leverages the OpenIDConnect and OAuth2.0 protocols and supports two types of users:

- Normal users: they can login to the INDIGO-IAM service, register client applications that use the INDIGO-IAM for authentication and link external accounts to their account;
- Administrators: they have administrative privileges within the INDIGO-IAM organization.

The INDIGO-IAM service exposes the OpenID Connect/OAuth dynamic client registration functionality provided by the MitreID OpenID Connect server libraries. By following the OAuth terminology, a client is an application or service able to interact with an authorization server for authentication/authorization purposes. Moreover, a new client can be registered in the INDIGO-IAM service using the dynamic client registration API. In the same way, the INDIGO-IAM dashboard can also be used for the same purposes. As soon as the client has been registered, it can obtain a token using both APIs or the available scripts and GUIs.

The INDIGO-IAM service has been successfully integrated with many off-the-shelf components such as Openstack, Kubernetes, Atlassian JIRA and Confluence and Grafana, and several middleware services that can be used in the IoTwins architecture.

### 5.2. Cloud Service Components

The management of computing resources available in the full stack, from IoT to edge and cloud, is entrusted to the IoTwins platform as a service (PaaS) orchestration

components [6]. Such components were developed in the context of the INDIGO-DC project [36] and were further extended within the already-mentined DEEP-HDC and XDC projects [37,38]. They are also in charge of checking the availability of the resources without interfering in the operation of the hosted computing infrastructure.

The IoTwins PaaS orchestration components follow the microservice-based architectural approach. The modularity of microservices makes the approach highly desirable for the architectural design of complex systems, specifically when many developers are involved. The orchestration component is further composed of a number of smaller sub-components, which are described hereafter, while details of the interactions among the sub-components are depicted in Figure 2.

- The **INDIGO Orchestrator** [39] is the core component of the PaaS orchestration. It accepts deployment requests expressed in OASIS's Topology and Orchestration Specification for Cloud Application (TOSCA) [40] format. From the received TOSCA-compliant request, the orchestrator implements a complex provisioning workflow aimed at fulfilling the request using information about the health status and capabilities of underlying IaaS and their resource availability, QoS/SLA constraints, the status of the data files and the storage resources needed by the service/application. This process allows for achieving the best allocation of the resources among multiple IaaS sites;
- The **Managed Service/Application (MSA)** deployment service [41] is in charge of scheduling, spawning, executing and monitoring applications and services on top of one or more Mesos clusters;
- The **Infrastructure Manager (IM)** [42] is in charge of deploying complex and customized virtual infrastructures on different IaaS cloud infrastructures. The IM eases the access and the usability of IaaS clouds by automating the VMI (virtual machine image) selection, deployment, configuration, software installation, monitoring and updating of the virtual infrastructure;
- The **Data Management Services**, a collection of services that provide an abstraction layer for accessing data storage in a unified and federated way, provide a unified view of the storage resources, along with the capabilities of importing data, and schedule transfers of data from different sources. The support of high-level storage requirements, such as flexible allocation of disk or tape storage space and support for the data life cycle, is achieved by the adoption of the INDIGO CDMI server [43]. The CDMI server has been extended in the INDIGO project to support quality-of-Ssrvice (QoS) and data life-cycle (DLC) operations for multiple storage back-ends, such as dCache [44], Ceph [45], IBM Spectrum Scale [46], TSM [47], StoRM [48] and HPSS [49];
- The **Monitoring Service** [50] is in charge of collecting and monitoring data from the targeted clouds and analyzing and transforming them into information to be consumed by the orchestrator;
- The **CloudProviderRanker** [51] is a rule-based engine that allows for managing the ranking of the resources eligible to fulfil the requested services. The engine uses the list of IaaS instances and their properties, provided by the orchestrator, to choose the best provider that can support the user requirements;
- The **Cloud Information Provider** [52] generates a representation of a site's cloud resources to be published inside the INDIGO Configuration Management Database (CMDB);
- The **Configuration Management Database** [53] is a REST service storing all the information about the cloud sites that is available and providing details such as the images and containers they support. It is used as an authoritative source of information for matchmaking and the orchestration of VM and containers;
- The **QoS/SLA Management Service** [54] allows for the handshake between users and resource providers on a given service-level agreement (SLA). The SLA describes the quality of service (QoS) for a group or a specific user, both in the PaaS as a whole or over a given provider.

*5.3. Edge Service Components*

The edge-layer software is aimed at providing services that are able to establish communication with both the IoT layer and the cloud layer. The applications in the IoTwins architecture are packed in OCI [55] containers. The edge service orchestration, thus, has to provide functionality to download, start, stop and configure the OCI containers.

In the IoTwins infrastructure, Apache Mesos [56] has been used as the service orchestrator in the edge. Apache Mesos is an open-source project to manage computer clusters and has two frameworks: Marathon [57], a production-grade container orchestration platform that can launch applications and provide scaling and self-healing for containerized workloads, and Chronos [58], a fault-tolerant scheduler that runs on top of Apache Mesos and that is used for job orchestration.

Within Mesos, docker containers can be launched using JSON definitions that specify the repository, resources, number of instances and command to execute. Scaling up can be accomplished by using the Marathon UI, and the Marathon scheduler will distribute these containers on slave nodes based on specified criteria. In Mesos, containers can be scheduled without constraints on node placement, or with constraints based on node types, e.g., one to one (the number of slave nodes should be at least equal to the number of containers). Mesos and Marathon support high availablity, which is implemented by the Zookeeper service. Zookeeper, in fact, provides the election of the Mesos and Marathon leaders and maintains the cluster state. Host ports can be mapped to multiple container ports, serving as a front end for other applications or end users. The status of the instances running on a docker container is continuously monitored by Marathon: in case one of the containers fails, Marathon is able to reschedule it. Auto-scaling is also supported using resource metrics available only through community-supported components. Local persistent volumes are supported for stateful applications such as MySQL. When needed, tasks can be restarted on the same node using the same volume. At the present time, applications that use external volumes can only be scaled to a single instance because a volume can only attach to a single task at a time.

Starting from the available Mesos frameworks, Marathon, which allows one to deploy and manage long-running services (LRSs), and Chronos, which allows one to execute computing tasks in a job-like fashion, the INDIGO-DC project has added new interesting functionalities in Mesos that make them suitable for exploitation: the elasticity of a Mesos cluster, so that it can automatically shrink or expand depending on the task queue; the automatic scaling of the user services running on top of the Mesos cluster; and a strong authentication mechanism based on OpenID-Connect. The INDIGO-DC components provide the needed level of integration among the cloud orchestration layer and the edge orchestration layer, enabling the PaaS orchestrator to deploy and make services available on the edge layer (see Figure 2). Moreover, within the IoTwins project activities, an all-in-one Mesos–edge cluster has been containerized and distributed among the partners who decide to host their own Mesos–edge that is directly connected with the orchestrator PaaS.

## 6. Use Case Scenario

In the present section, the use case proposed by CETIM [59] is presented and discussed, together with the computing services used by the testbed and made available via the IoTwins platform.

CETIM is a Technological Institute of Mechanics steered by mechanical engineering industrialists under French State supervision. As regional, national and international player, they are providing much support to SMEs in several domains of physics and mechanics. The commercial offer covers the following domains:

1.  Measurements and testing;
2.  Expert analysis;
3.  Consultancy and training;
4.  Systems engineering;
5.  Innovation.

The testbed will explore how technological inputs provided by the IoTwins project will improve the SMEs' competitiveness in terms of production performance, component quality and maintenance cost. In particular, it will prove the ability to develop a generalized and replicable smart manufacturing methodology for SMEs. The developed distributed DTs, powered with AI operating on collected operational data and augmented by heavyweight simulation runs, will enable production performance improvements in terms of component quality and tolerance, thanks to the adequate real-time decision support offered by the IoTwins infrastructure (with respect to the latency constraints). It is important to mention that the distributed DTs developed within this testbed are specific for each machine and can evolve over time.

Three kinds of RT will be explored:

-   Model-driven twin, essentially based on simulation tools, that will provide decision support to product planning departments on how to explore optimal ways of implementing production;
-   Data-driven twin, built on top of sensor data. Basically, it consists of a ML model trained with historical data in the cloud and then used in the edge to monitor production;
-   Hybrid twin, which combines the benefits of the model-driven and data-driven twins. The MOR-based simulator used in online operations shows a very low response time, which makes it compatible with the real-time constraints of the process. In spite of this, the gap between the simulated and real values is still unacceptable. The output of the data-driven model is then used to support the simulator designers to fill that gap.

### 6.1. Testbed Description and Initial Requirements

The testbed will evaluate the digitization of manufacturing processes following the Industry 4.0 recommendations [60]. This use case focuses on the manufacturing of parts through turning and milling processes. **Turning** is a machining process for manufacturing parts by the combination of two movements: a rotational movement is applied to the part while the tool is in translation to result in chip removal. **Milling** is a machining process in which the tool (the cutter) undergoes a rotational movement. The advance of the part by a translational movement then makes it possible to eliminate the chips.

The "as-is" situation, before joining the IoTwins project, was as follows:

-   Simulation capabilities were missing. A machining simulation software able to easily compute forces and spindle power would be a plus. Output of simulation could be used to define machine features and limitations;
-   During the process setup (pre-production phase), the operator requires the support of software advisors and data analysis tools. The objective is to reduce the new production setup time from 1/2 day to 30 min, and to improve the trial/error phase;
-   Need of an optimized operator schedule (calling the operator only when needed for tool changing, typically scheduling the control of the machining process every two hours instead of every hour);
-   Need of process monitoring and capitalizing on the events during production time;
-   Improvements of part quality (reduce the production of bad parts by detection/prediction using sensor signals and models);
-   Need for more accurate and improvable predictions over time (capitalization and fine-tuning of prediction models).

After an accurate analysis of the needs, the "to-be" scenario was developed, which included the integration with the IoTwins platform. Figure 3 shows the data flow and the

operations required to implement the new scenario, and the infrastructure layers (edge and cloud) involved in the computing steps.

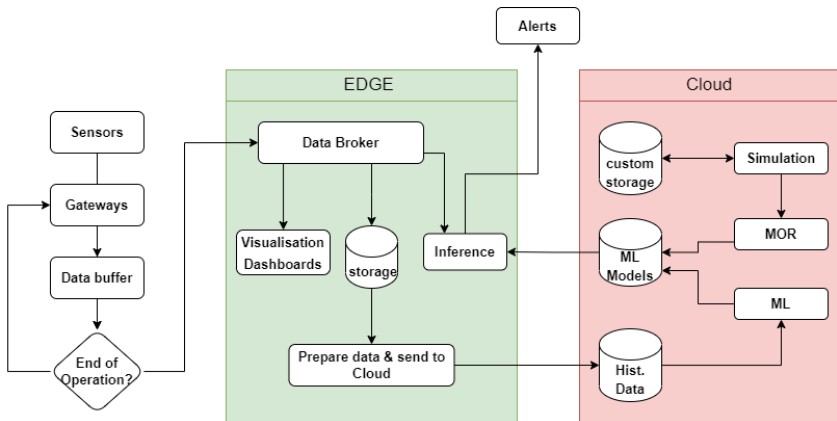

**Figure 3.** The "to-be" scenario: data flow describing the required operations and the involved computing infrastructure.

Figure 3 is a typical representation of the data stream as well as the functions that apply to them. The shop floor machines are equipped with several sensors to measure forces, acceleration, bar feeder vibrations, surface roughness, acoustic emissions and lubricant temperature and pressure. During an operation, each sensor transmits data to an acquisition box that serves as an IoT Gateway. The latter buffers the data until the end of the operation. The data packet is then transmitted to the edge, where a data broker will take care of dispatching it to various applications:

- Visualization of data;
- Local storage;
- Monitoring and follow-up of production (later on, "notifier"), whose purpose is to issue alerts when necessary.

From here, raw data will be periodically pre-processed and sent to the cloud, where they are stored as historical data. On the cloud end, the user will access resources for simulation, ML and model order reduction (MOR) services, which are fed with the above-mentioned historical data.

### 6.2. Requirements of the IoTwins Platform

The projected IoT solution aims at digitizing manufacturing activities. Figure 4 presents a component-based view of the implemented system. The latter was built with the support of the tools provided by the IoTwins platform and covers both the edge and cloud layers. At present, the IoT layer only streams data to the edge layer.

Different software tools are required to fulfil the needs of the testbed application. In order to fulfil the user needs, platform components, proprietary components and commercial components have been combined and deployed. A subset of such components, also illustrated in Figure 4, are discussed below. The platform provides the CETIM with many dockerized services. Some of them helped in building the targeted solution for the CETIM's use case.

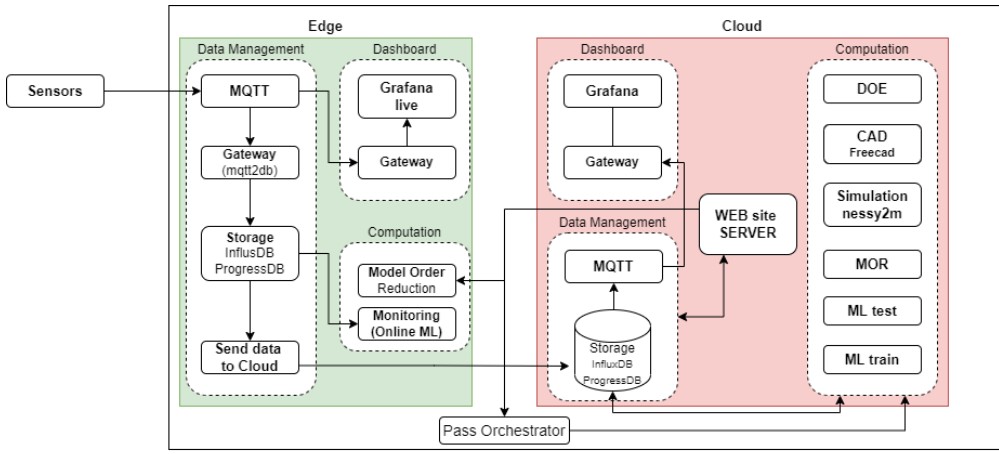

**Figure 4.** Component-based view of the implemented solution.

- **Edge computing environment**: edge resources are virtualized by means of the software environment provided by the IoTwins edge service runtime. The latter hosts a Mesos cluster to run jobs as containers, which are managed and orchestrated by the Marathon and Chronos frameworks. All containers are hosted at Harbor [61], the open-source registry made available by the IoTwins platform to the IoTwins testbeds;
- **Communication**: in both edge and cloud, RabbitMQ [62] is used. MQTT [63] is the communication protocol implemented to manage the data streaming between IOT and edge, to send data to the cloud and to manage notifications;
- **Data Storage**: several storage types are used to manage data storage. On edge, two database systems are used for the local storage, Postgres DB [64], an SQL database and InfluxDB [65], which contain the time-series data issued by the sensors;
- **Visualization**: data visualization is performed through **Grafana** [66], which displays live sensor data fetched from the MQTT broker deployed at the edge machine. Additional dashboards, developed in the scope of the IoTwins projects, display views combining raw data and training model information. The simulation also requires specific tools to manage visualizations of the results;
- **Simulation**: the simulation capabilities for machining processes virtually simulate the reality and provide quantitative and qualitative information about the processes and the machine parts. Several decisions may be impacted by such information. Power consumption information may be used to select the most appropriate production machine. Forces predict the the behavior of the tools during the part fabrication and may be used to anticipate breaks and failures. Vibrations are correlated to the tools' wear. Altogether, this information supports the user to design the process, set the process parameters and define the production strategy;
- **ML training**: data analytics is deployed to detect the default in a machining process. For simplicity, the parts are labelled as damaged (1) or good (0). According to the level of error, three zones are defined to access the quality of the part and an appropriate alert is notified: green zone (good parts), when the error value is 2%, orange zone (average parts), when the error value is between 2% and 5%; and red zone (bad parts), when the error value is above 5% (percentages are expressed in absolute values). The first machined parts are used to extract the features (modes), via the proper orthogonal decomposition [67] (POD) methodology, which is similar to the singular value decomposition (SVD) method, of what are qualified as "good parts". The extraction of the data is performed on the sensors' data, measured on the tool. The forces of the tool are used as indicators of the state of the machined part. Trained models (modes), built offline and stored, will be sent to the edge machine in the production phase. Measured sensor data are projected on the (stored) modes (describing the good parts).

The projection error helps to define the state of the current machined part. According to the level of error, an alert notification may be issued.

### 6.3. UML Sequence Diagram of a Digital Twin Deployment

By leveraging the functionalities offered by the IoTwins Platform, CETIM implemented an automated deployment of the digital twin (DT). In this section, we provide a detailed description of the steps taken to achieve the deployment task, making use of the services provided by the platform. For space reasons, only the description of the data-driven DT is provided.

In Figure 5, an UML-compliant sequence diagram of the deployment is depicted. The process is triggered by the IoTwins user, who feeds the INDIGO-DC PaaS orchestrator, one of the IoTwins PaaS components, with a TOSCA-compliant blueprint describing the DT topology and the application requirements. The PaaS orchestrator component parses the blueprint and transforms the instructions into a workflow of actions that are executed with the support of a workflow engine.

As a first step, a docker-enabled virtual machine has been deployed on the INFN cloud infrastructure and made available for the testbed via the PaaS orchestrator. The following steps include, sequentially, the provisioning of a MinIO instance and of a pre-designed ML model. Both of the instances have been dockerized and run as containers. A further configuration step (not shown in Figure 5) is taken to configure the ML model to fetch data from the MioIO instance. After the deployment succeeds, web endpoints of the above-described services are returned through the PaaS orchestrator. This last step concludes the provisioning/configuration of the piece of the DT hosted in the cloud.

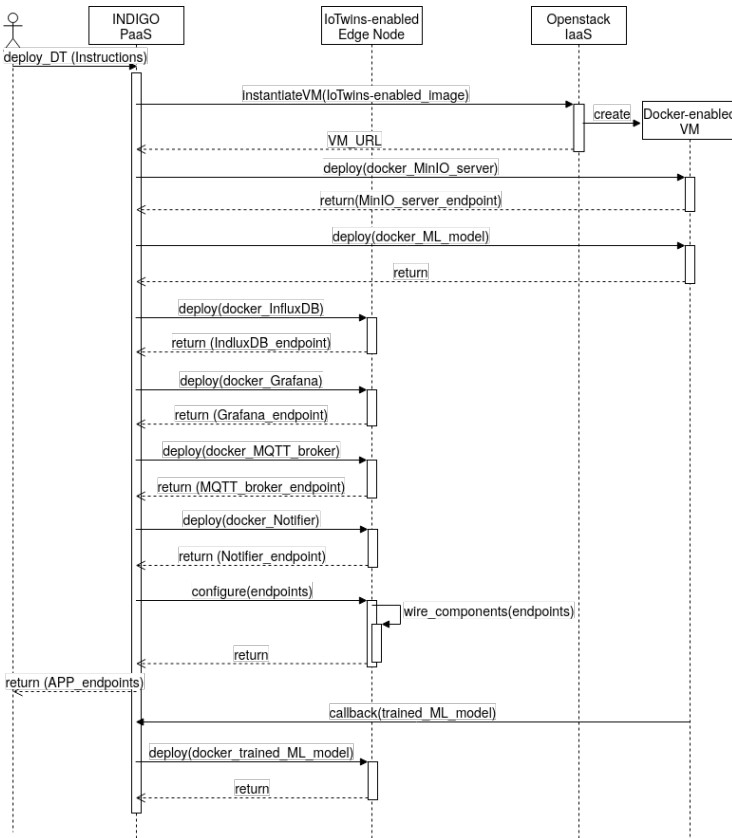

**Figure 5.** Digital twin deployment steps, starting from the user submission template to the return of the trained ML model from the edge and cloud layers.

The following actions are then carried out to implement the provisioning of the edge-hosted DT. The latter is composed of the following docker components: an InfluxDB storage, a Grafana web application, an MQTT broker and a web-based notifier application. The

MQTT broker provides the data stream that feeds (among others) the InfluxDB, which, in turn, will feed the dashboards implemented in Grafana. A final configuration step will take care of wiring all deployed components. Since all software components are offered as docker services (offering a REST interface), the sequence of the deployment steps is not relevant, as long as the final configuration step sets the service pipeline up correctly. In the figure, one possible sequence is depicted. Finally, the endpoints of the Grafana and the notifier web applications, respectively, are returned to the IoTwins user.

Immediately after its deployment in the cloud, the ML model is trained with input data fetched from the MinIO storage. Once the training is completed, the model is shipped back to the PaaS orchestrator via an asynchronous callback. Then, the PaaS orchestrator shifts the model to the edge, where it is fed with a real-time data stream produced by the field instrumentation; in case of anomalies, it will trigger alarms to the notifier application.

We stress that the MQTT broker deployed in the edge provides data streams to (i) the InfluxDB storage residing in the edge, (ii) the trained ML model residing in the edge and (iii) the MinIO instance residing in the cloud.

### 6.4. The Application and Development Context

In the current implementation, a solution proposed to the final user includes a front-end component, a production monitoring tool and a back-end component.

The *front-end* is a web application that provides the user with several capabilities, covering the product panning concerns as well as the ones related to the production monitoring. This will highlight the application that will support the IOT solution. On the one hand, the software tools necessary for the implementation of the digital twin are offered. They are articulated around the simulation of machining. This allows for the preparation of the data, the definition of the problem and the post-processing of the results. An example is illustrated in Figure 6, showing the simulation results as curve. Power consumption, torque and cutting forces are presented, and can be selectively displayed upon the user's choices.

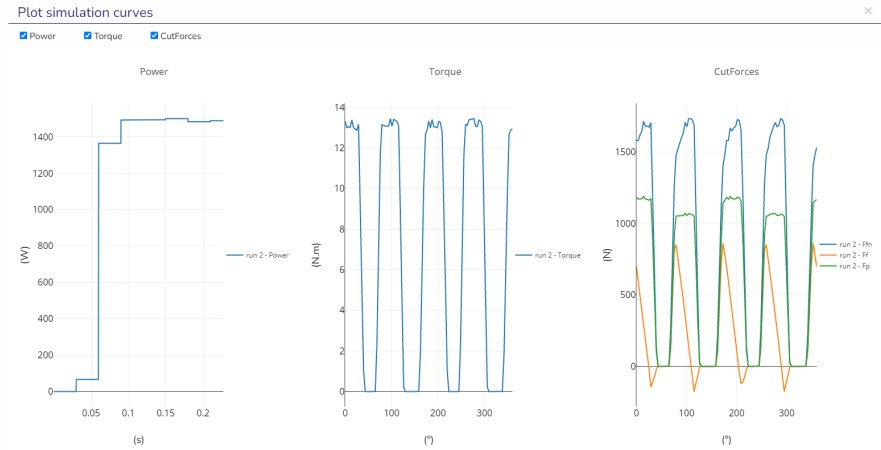

**Figure 6.** Simulation results visualization: power consumption, torque and cutting force trends.

The outcome of this simulation phase is used in the product planning department to:

- build the sequence manufacturing process operations;
- identify potential issues that may come during the production phase;
- select the most appropriates machines for the production based on the computed forces and the machine power.

On the other hand, an ML-based *production monitoring tool* is proposed. Based on a sensor data subset, a digital twin model was built. By comparing the measured values on the one hand and the calculated values on the other, it is possible to deduce errors specific to each operation in the form of a time series. The characterization of this error will define a status (good, average, bad) for the current operation. Figure 7 illustrates the use of this

modeling for the forces. On the top side of the image, the model built during the training phase is displayed. The blue curve shows the raw signal. The red curve superposed to it represents the model curve. Below, the error between the raw and the predicted data is drawn. This error will serve to access the quality of the manufactured part. On the bottom end of the picture, we outlined the model in action. The next step will be to bring the simulation and the ML closer together to build the hybrid twin. The modus operandi would be to complete the simulation by the ML to model the gap between the simulated data and the data captured by the sensors. This composite model would, thus, be more realistic in order to monitor the production processes.

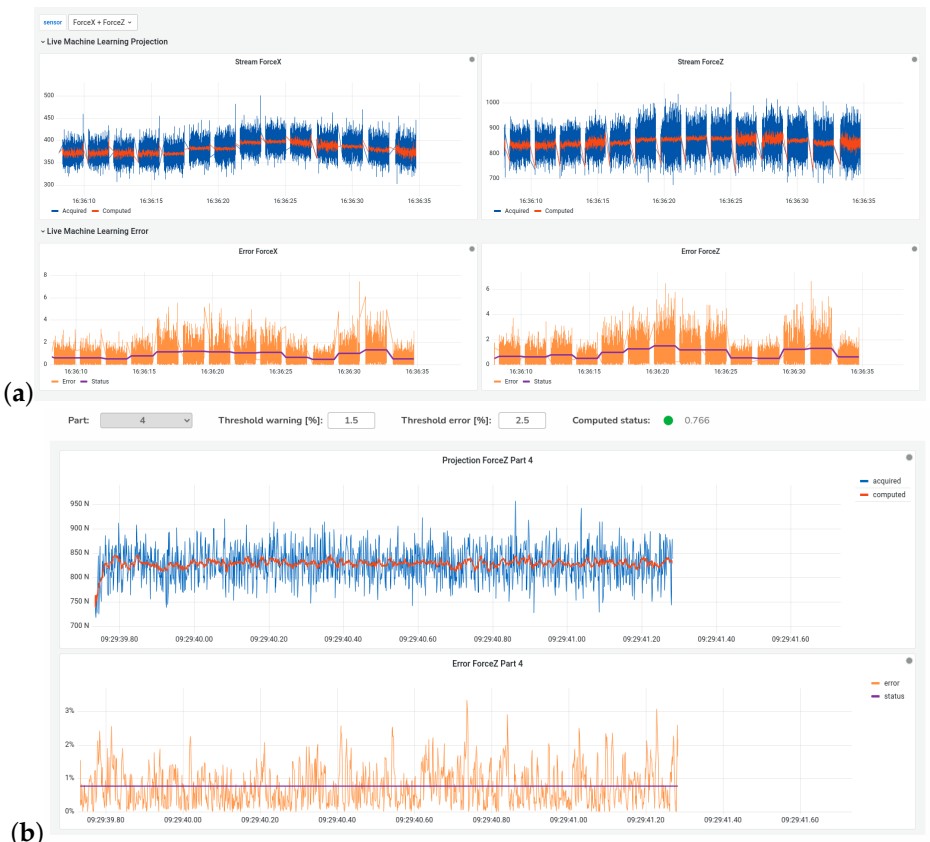

**Figure 7.** ML model: training (**a**) and execution (**b**).

Finally, the *back-end component* exploits the IoTwins functionalities and enables the deployment of the cloud/edge/IoT services provided by the platform and specifically developed for the testbed. The IoTwins platform provides different services aimed at best exploiting the cloud–edge–IoT layers and offering the needed support to achieve the results. The openness of the IoTwins platform allows the user/developer to use their own software artefacts, integrate them within the edge and cloud context and accelerate the adoption of digital twin techniques.

## 7. Conclusions and Future Work

This paper reports an effort made to define and deploy a set of software services to be proposed as an highly distributed computing platform for a hybrid digital twin model, providing integration of simulative and data-driven models to feed AI services.

The project followed the use case-driven approach in order to collect the different requirements and, accordingly, to select and implement a set of functionalities aimed at supporting the final user with:

- data transfer to/from the different infrastructure levels;
- data types to accommodate different data storage needs;

- elaboration of data both for streamed-data and batch-data elaboration.

Furthermore, following a proven software development approach, the requirement collection and analysis phase was followed by applying techniques aimed at defining a general high-level platform that has been presented and described. For validation purposes, a complete industry-related use case, showcasing different computing services accessible through the IoTwins platform, has also been presented.

The present work lays the foundation to build a production-type platform for hybrid digital twin models' calculation and evaluation. IoTwins claims that IoT, edge computing and industrial cloud technologies, together, are the cornerstones of the creation of distributed digital twin infrastructures that, after testbed experimentation, refinement and maturity improvements, can be easily adopted by SMEs. The experience and discussions emerging from the project activities will directly influence further developments aimed at accelerating the adoption of digital twin techniques by exploiting SMEs' industry-perceived advantages in terms of increased reliability/autonomy and of improved locality preservation of critical production data that can be maintained and used directly in a plant's premises.

The platform is currently being tested on other manufacturing testbeds within the IoTwins project. Test results are used to tune the platform specifications and implement new functionalities. Specifically, activities aimed at supporting other orchestration frameworks in the edge (e.g., Kubernetes [68]) have been undertaken. With the aim of delivering a ready and easy-to-use product to the manufacturing community, the following points still need to be addressed: (i) improving platform scalability to accommodate a large number of users; (ii) improving the usability of front-end tools; (iii) adding more data services to enhance compliance with other data management tools; (iv) increasing platform robustness by running massive functional tests. Some of these points (adding data services, improving scalability) will be addressed in the course of the project, others are part of long-term future work.

**Author Contributions:** Funding acquisition, D.C.; Investigation, B.M., M.G., M.A., D.N., P.B., C.D. and D.C.; Supervision, A.C.; Writing—review & editing, A.C., G.D.M., J.C.A. and D.C.D. All authors have read and agreed to the published version of the manuscript.

**Funding:** This work was partially supported by the EU H2020 IoTwins Innovation Action project (g.a. 857191).

**Institutional Review Board Statement:** Not applicable.

**Informed Consent Statement:** Not applicable.

**Data Availability Statement:** Data has been present in main text.

**Conflicts of Interest:** The authors declare no conflict of interest.

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
