# Peer review of "IoTwins: Toward Implementation of Distributed Digital Twins in Industry 4.0 Settings"

_computers, doi:10.3390/computers11050067_

Round 1
Reviewer 1 Report
The authors present a very practical use of the digital twin paradigm in the manufacturing domain. The presentation is clear, bibliography is relevant, the conclusions are supported by the validation. The article is of interest to the readers as it describes the entire lifecycle of the implementation of a DT-based system, and it showcases the advantages of the hybrid approach.
My only recommendations are:
-better proofreading, some references are not correctly displayed
-describe the actual machine learning algorithm used to simulate the model
Author Response
Let me take this opportunity to thank the Reviewer 1 for their useful comments. As requested by the Editor, a point-by-point response has been provided. To simplify the referees’ work, main modifications have been highlighted in red color in the manuscript.
Comment 1.1: better proofreading, some references are not correctly displayed
Response 1.1: Text integrations and adjustments have been added into the manuscript to improve readability. Moreover, a deep check of references and punctuations has been done as pointed out by the reviewer.
Comment 1.2: describe the actual machine learning algorithm used to simulate the model
Response 1.2: We provided a better description of the Machine learning algorithm employed in the testbed (see Section 6.2, last item of the bullet-list)
Reviewer 2 Report
The paper discloses the details of the IoTwins project, a funded project under the EU Innovation Actions framworkk. The paper aims to disclose the project development, namely by presenting the architecture framework, detail on the platform components and a use case highlighting the applicability and readiness of the platform and its functionalities.
The paper is well structured and is easy to read. Some minor errors where noticed as it will be detailed further.
The project main purpose is to democratize Industry 4.0 to companies that do not have the capability to invest in tools as this one from scratch.
The novelty and contribution for the body of knowledge relies on the detailed explanation of the assumptions, used technologies and how they suit the purpose by presenting a practical use case.
Despite the research presented, this paper relevance is on the developments and on the use case, leading this to a very practical manuscript.
However, there are some aspects that could be improved and the following part addresses these:
- Section 2, lines 73 to 75. I would like further comment on this sentence as I understand that DT is most of the times set up “copying” the physical element. However, would you consider that it might born even before the physical element, meaning as part of the design process of the physical counterpart?
- Line 82, reference 12 is missing.
- Around line 200. I understand the use of existing technologies but this should be better explained, namely the Why? to choose one instead of other. Further explanation on this would improve the paper quality and allow readers to understand more clearly the “pros and cons” of the platform components.
- Section 6.1 – it would be interesting to add a figure presenting the required operations before. This would allow an improved perception of the changes and on the potential gains.
- Line 433 – Figure 4 is missing. If the previous comment is considered this will become Figure 5.
Author Response
Let me take this opportunity to thank the Reviewer 2 for their useful comments. As requested by the Editor, a point-by-point response has been provided. To simplify the referees’ work, main modifications have been highlighted in red color in the manuscript.
Comment 2.1: Section 2, lines 73 to 75. I would like further comment on this sentence as I understand that DT is most of the times set up “copying” the physical element. However, would you consider that it might born even before the physical element, meaning as part of the design process of the physical counterpart?
Response 2.1: The comment is correct, and we thank the reviewer for pointing it out. The activities carried out in the present work are based on the use cases recruited for the IoTwins project. Those use cases are centered on the definition and the development of DT replicas to reproduce the existing physical environment and perform on that the selected ML techniques and related studies.
Comment 2.2: Line 82, reference 12 is missing.
Response 2.2: A deep check of the references, their content and related position in the text has been performed and errors have been corrected where needed.
Comment 2.3: Around line 200. I understand the use of existing technologies but this should be better explained, namely the Why? to choose one instead of other. Further explanation on this would improve the paper quality and allow readers to understand more clearly the “pros and cons” of the platform components.
Response 2.3: We thank the reviewer for the comment. The technologies adopted and used have been selected from previous projects. The criteria used to select the technologies and related tools were: open source software, in order to avoid the vendor lock-in; the support level, the communities that contribute to the development of the respectives tools and technologies; the level of standards adoption, de jure and de-facto". A sentence has been added in Section 4 in order to clarify that aspect.
Comment 2.4: Section 6.1 – it would be interesting to add a figure presenting the required operations before. This would allow an improved perception of the changes and on the potential gains.
Response 2.4: In our opinion, the description of requirements is best rendered through text as this allowed us to provide more details than a picture would do. That is why we prefer to keep the actual representation of requirements as it is. Please, consider also that in the “to-be” picture, we are depicting a new data flow that in the “as-is” architecture is not present.
Comment 2.5: Line 433 – Figure 4 is missing. If the previous comment is considered this will become Figure 5.
Response 2.5: Figures and related references in the text have been checked. A typo in the label of Figure 4 has been found and corrected.
Reviewer 3 Report
In my opinion, the manuscript computers-1659155 can be published in the Computers journal after major revisions.
Details of my revision are in the attached file.

Author Response
Let me take this opportunity to thank the Reviewer 3 for their useful comments. As requested by the Editor, a point-by-point response has been provided. To simplify the referees’ work, main modifications have been highlighted in red color in the manuscript.
Response 3.1: We carefully reviewed the paper to address the following suggestions pointed out by the reviewer:
- References have been deeply checked and formatted as from the Journal guidelines;
- Tables and figures have been formatted as from the Journal guidelines;
- Punctuations and bullet lists have been checked and errors have been corrected;
- Details have been added in the Authors' affiliations

Reviewer 4 Report
Dear authors,
I really thank you for giving me the possibility to review the paper named : "IoTwins: toward implementation of distributed digital twins in Industry 4.0 settings". I hope you and your families are safe and well.
Looking at your paper I find it very interesting. The topic is important and quite well structured.
Going to the paper :
a) Introduction - related work -I would improve better the part of I4.0 that is marginal in the general context. I suggest improving with more literature for the context. The paper is well- written but I have some difficulities to understand quickly the topic. Then in the other sections it clarifies well. I don't suggest any paper now, it's not my cup of tea.
b) about conclusions : I suggest to put into evidence better the weakness of the work and the future research.
c) from a plagiarism check it is too high. Please check it
Author Response
Let me take this opportunity to thank Reviewer 4 for their useful comments.
As requested by the Editor, a point-by-point response has been provided.
To simplify the referees’ work, main modifications have been highlighted in red color in the manuscript.
Comment 4.1: a) Introduction - related work -I would improve better the part of I4.0 that is marginal in the general context. I suggest improving with more literature for the context. The paper is well- written but I have some difficulties to understand quickly the topic. Then in the other sections it clarifies well. I don't suggest any paper now, it's not my cup of tea.
Response 4.1: The Introduction now contains a paragraph on Industry 4.0 that accommodates the reviewer’s comment. We also enhanced the literature review section with relevant works on I4.0 that better depict the background of our work.
Comment 4.2: b) about conclusions : I suggest to put into evidence better the weakness of the work and the future research.
Response 4.2: We thank the reviewer for the comment. In the conclusive section of the paper, we highlighted the major flaws of the platform. Some of those will be addressed in the course of the project, others will be addressed in the long term.
Comment 4.3: c) from a plagiarism check it is too high. Please check it
Response 4.3: Thank you for pointing this out. The Editor provided us with the report and we worked to mitigate it by paraphrasing the text. Modifications have been highlighted in blue color in the manuscript.

Round 2
Reviewer 3 Report
Accept in present form